# Physical Modeling of Ski-Jump Spillway to Evaluate Dynamic Pressure

**Mehdi Karami Moghadam [1], Ata Amini [2,]\* , Marlinda Abdul Malek [3], Thamer Mohammad [4] and Hasan Hoseini [5]**

1   Department of Agriculture, Payame Noor University (PNU), Tehran 19395-4697, Iran
2   Kurdistan Agricultural and Natural Resources Research and Education Center, AREEO, Sanandaj 6616949688, Iran
3   Institute of Sustainable Energy, Universiti Tenaga Nasional, Selangor 43000, Malaysia
4   Department of Water Resources Engineering, College of Engineering, University of Baghdad, Baghdad 10011, Iraq
5   Faculty of Water Sciences Engineering, Shahid Chamran University of Ahvaz, Ahvaz 6135783151, Iran
*   Correspondence: a.amini@areeo.ac.ir or ata_amini@yahoo.com; Tel.: +98-91-8371-4538

**Abstract:** The effects of changes in the angle of pool impact plate, plunging depth, and discharge upon the dynamic pressure caused by ski-jump buckets were investigated in the laboratory. Four impact plate angles and four plunging depths were used. Discharges of 67, 86, 161, and 184 L/s were chosen. For any discharge, plunging depth and impact plate angle were regulated, and dynamic pressures were measured by a transducer. The results showed that with the increase in the ratio of drop length of the jet to its break-up length ($H/L_b$), and with an increase in the impact plate angle, the mean dynamic pressure coefficient decreased. An inspection of the plunging depth ($Y$) ratio to the initial thickness of the jet ($B_j$) revealed that when $Y/B_j > 3$, the plunging depth of the downstream pool reduced dynamic pressure. At the angle of 60°, the dynamic pressure coefficient due to increasing in plunging depth varied from 34% to 95%.

**Keywords:** jet falling; energy dissipation; surface disturbances; pressure fluctuations; water jet; physical modeling

## 1. Introduction

Owing to dynamic pressures resulting from the flow in hydraulic structures, the river bed is frequently affected with scouring [1,2]. To dissipate the flow energy and to avoid this scouring, dissipater structures such as a spillway with a fillip bucket, which is applied at the end of chute spillway, are used [3]. The flow in the structure is thrown into the air by a ski-jump and goes down after dissipation of part of the energy. Steiner et al. [4] conducted experiments on triangular jets, and compared parameters such as dynamic pressure over the bucket, as well as energy dissipation between triangular and circular buckets. Their results indicated that the relative energy dissipation hinges on the deflection angle and jet falling height from the take-off lip to tail water level. Jorabloo et al. [5] simulated the ski-jump stream outlet through the Fluent model and compared the model output with the results of the physical model. They concluded that pressure distribution, as well as jet trajectory in the two models, are close to each other. Turbulent jet into the flow was numerically analyzed by Mahmoud et al. [6], along with studying the recirculation bubbles in the flow. Their results show that the size and power of the recirculation bubbles increase with the enlargement of the nozzle size. Furthermore, the bubbles disappeared as the Froude number was reduced. Chakravarti et al. [7] have investigated the static and dynamic scouring caused by submerged circular vertical jets. They

verified that the depth of dynamic scouring is greater than that of static scouring. Artificial neural network (ANN) was employed by Noori and Hooshyripor [8] based on the major effects of the input parameters on the downstream scouring of the fillip bucket. Their results showed that the Log–Sigmaid model had good performance in the modeling of the depth of scouring. The smooth particle hydrodynamics technique was adopted to study pressure distribution on the steps of a stepped spillway by Husain et al. [9]. Their results showed good consistency with the laboratory observations. Distribution of hydrostatic and non-hydrostatic pressure in shallow waters was investigated by Arico and Re [10]. Dividing the whole pressure into dynamic and hydrostatic components, they solved a hydrostatic and a non-hydrostatic problem sequentially in a fractional time step procedure. Simpiger and Bhalera [11] developed an equation for measuring the jet length according to the jet trajectory. They compared the obtained length with the observation values in a physical model and used the equation to calculate the jet trajectory length in the prototype. Aminoroayaie Yamini et al. [12] investigated the pressure fluctuations and the effect of the entering flow on the fillip bucket bed of Gotvand Dam in Iran. The results show that when the depth and discharge of the entering flow increases and Froude number decreases, the mean dynamic pressure declines and pressure fluctuations grow. They observed that the average pressure was at a maximum at the bucket entry, and was at a minimum at the end part of it. Wu et al. [13] studied the energy dissipation in a fillip bucket of slot-type, both numerically and in the laboratory, and suggested equations to estimate the energy dissipation value. The results showed that as the flow drops in downstream submerged or unsubmerged pools, the resulting dynamical pressure was transformed to the bottom and sidewalls.

Understanding the jet features is crucial in designing the pool and determining the plunging rate. Notwithstanding the research carried out in this regard, it seems that realizing the precise mechanism of ski-jet impacting on the pool requires more studies. In this research, the effects of the main variables upon changes in dynamical pressure and jet break-up length were investigated using the laboratory model of the spillway and the fillip bucket of a dam constructed in Iran. Comprehensive data, as well as the analyses presented in this study, can be used by researchers and engineers.

## 2. Materials and Methods

### 2.1. Mechanism and Effective Parameters

The effective parameters in jet break-up include fluid properties, the environment features, and the jet outlet conditions. A schematic drawing of regions and parameters in a jet break-up is depicted in Figure 1.

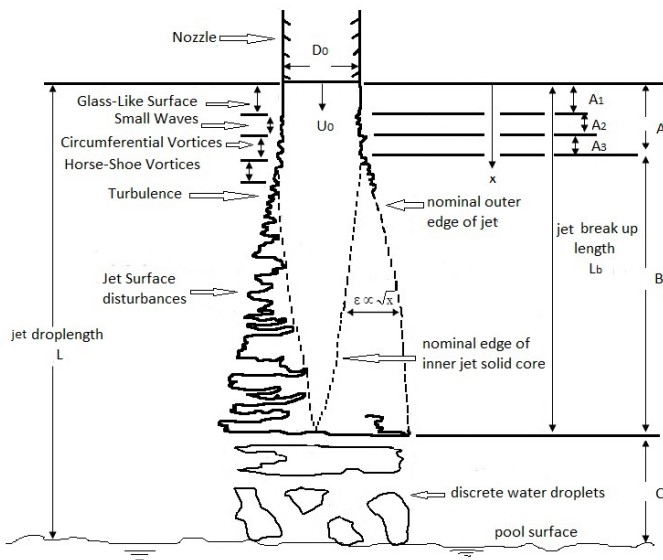

**Figure 1.** Features of falling jet and its development (adapted from Ervine et al. [14]).

As shown in Figure 1, there are three flow regimes to be considered before the vertical jet impacts the water surface. Region A is composed of three sub-regions: $A_1$, $A_2$, and $A_3$. In subregion $A_1$, when the flow exits the nozzle, the surface tension resists disturbance, meaning the jet surface remains flat and glass-like. In subregion $A_2$, roughness (waves) grows at the water surface. As for subregion $A_3$, surface waves turn into circumferential vortices. Region B is where surface disturbances ($\varepsilon$) increase with the square root of the fall distance ($\varepsilon \propto \sqrt{X}$), where X is the distance from the beginning of the jet. The air penetrates the jet perpendicular to its trajectory. The distance from the jet beginning up to the end of region B is called break-up length ($L_b$). Very intense surface disturbances enter the jet in region C, whereby the flow gets out of continuity mode. In region C, the flow is not of continuous mass and the flow masses are quite distinct. Surface tension and turbulence determine the distance of $L_b$, where the jet breaking-up occurs and causes the jet impacts with less energy (Ervine et al. [14]). The jet along its direction may be either core or non-core. In Figure 1, up till region B, the core jet exists, and in region C, there is the non-core situation. In case the plunging is sufficient, the non-core jet expands to the sides as vortices.

### 2.2. Dimensional Analysis

The parameters governing the dynamic pressure resulting from impacting the fillip bucket jet on the plunge pool floor are expressible as in Equation (1) [15]:

$$f(q, \rho_w, \Delta P_{max}, g, B_j, Y, H, R, \alpha, \varphi, \mu, H, L_b) = 0. \tag{1}$$

In Equation (1), q stands for discharge per unit width, $\rho_w$ is water density, $\Delta P_{max}$ represents the maximum pressure head, g is the gravitational acceleration, $B_j$ designates impingement jet thickness, Y is tail water depth, H is the drop height, R is radius of fillip bucket, $\alpha$ signifies the angle of impact plate with horizon, $\varphi$ is the take-off angle of jet with horizon, $\mu$ stands for water dynamic viscosity, H is the jet length, and $L_b$ is the jet break-up length. The dimensionless parameters may be represented, through dimensional analysis, as Equation (2) [15]:

$$f(Re, Fr, \frac{\Delta P_{max}}{Y}, \frac{H}{Y}, \frac{R}{Y}, \frac{B_j}{Y}, \alpha, \varphi, \frac{H}{Y}, \frac{L_b}{Y}) = 0, \tag{2}$$

where Re stands for Reynold's number and Fr is Froude number. By removing the fixed parameters, the average dynamic pressure coefficient ($C_p$) is obtained as follows:

$$C_P = \frac{H_m - Y}{\frac{U_j^2}{2g}} = f(\alpha, \frac{H}{L_b}, \frac{Y}{B_j}), \tag{3}$$

where $H_m$ is the average of observed dynamic pressures, and $U_j$ designates the jet velocity at the impingement moment. The break-up length is calculated from Equations (4) and (5) [16]:

$$\frac{L_b}{B_i Fr_i^2} = \frac{0.85}{(1.07 T_u Fr_i^2)^{0.82}}, \tag{4}$$

$$T_u = \frac{RMS(u')}{\bar{u}}, \tag{5}$$

where $B_i$, $T_u$, and $Fr_i$ are initial thickness, turbulence intensity, and initial Froude number of the jet, respectively. RMS ($u'$) is the root mean square of the values of the velocity fluctuations in the cross section and in the direction of the falling jet axis, and $\bar{u}$ is the average jet velocity.

### 2.3. The Experiments and Models

The experiments of the present research were conducted in the hydraulic laboratory of Chamran University, Ahwaz, Iran. The laboratory models of the spillway and fillip bucket were scaled down of

Balarood Dam spillway, located 27 km north of Andimeshk, Khouzestan Province. Balarood dam is constructed on the Balarood River and is of the earthy type with clay core. It is 75.5 m in height and 1070 m crest length, and has a reservoir volume of 131 MCM. The laboratory model of the spillway was scaled down with a scale of 1:40 based on the principle of dynamic simulation, and with due regard to the flume and discharge conditions in the laboratory. The dimensions of the flume were 0.5 m width, 9 m length, and 2 m height. Four discharges of 67, 86, 161, and 184 L/s were taken corresponding with the real discharges of 674.85, 870, 1622.2, and 1857.2 m³/s in the prototype, respectively. The latter discharges correspond to those with the return periods of 2, 100, 1000, and 5000 years. The four downstream water depths used in the plunge pool were 0, 15, 30, and 45 cm. Also, angles of 0°, 30°, 60°, and 90° were chosen for the impact plate in the plunge pool. Taking these variables into account, a total of 64 experiments were carried out.

## 2.4. Experiments' Setup

During the experiments, the flow was established by a pump using a circulation system. To minimize turbulence, stilling reservoir and honeycomb were utilized before the flow reaches the experiment area. At the beginning of each experiment, the required discharge was regulated via a gate valve and a rectangular weir installed at the end of the flume. Figure 2 shows a schema of the ski-jump jet (Figure 2a) plus parts of the physical model (Figure 2b). A Plexiglass 0.5 m × 0.5 m square plate was employed in designing the impact plate on which the jet arising from the fillip bucket was to be impinged. A total of 37 pores, each with a diameter of 2 mm, was set on the impact plate in order for connection of the piezometer tubes. The impact plate was placed on a metal system movable in a vertical direction in order to set particular water depth at the position of jet impingement. Also, the impact plate was able to rotate around the horizontal axis to create various angles with the horizon. By moving and rotating, the impact plate angle ($\alpha$) and the plunging depth (Y) were adjusted as in Figure 2a. Finally, the flow was discharged through the return channel system into the primary reservoir.

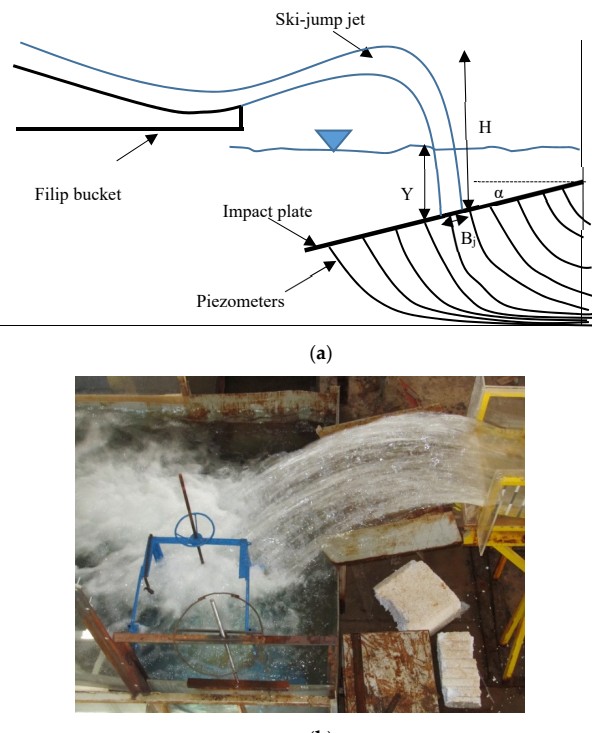

(**a**)

(**b**)

**Figure 2.** (**a**) Ski-jump, plunging depth, and angle of impact plate; (**b**) jet trajectory of ski-jump and impinge on impact plate.

### 2.5. Measurement of Dynamic Pressure

Piezometers were used to observe the fluctuations of the dynamic pressures on the impingement place. Those connected to the impact plate were stretched out from the bottom of the reservoir wall and moved to the dial board, as shown in Figure 3. After the flow exhausted from the bucket and the jet impinged on the impact plate, the dynamic pressures were measured. To demonstrate the dynamic pressures, two of the 37 piezometers, showing the highest pressures, were linked to a transducer. The transducer converts the dynamic pressure of the piezometers into electrical signals and transmitted to the amplifiers by the particular cables. For 10 min, 50 data of dynamic pressures per second were taken. The accuracy of the transducer was ±1 mm for the laboratory model corresponding to 0.04 m for the prototype (Balarood dam). The data obtained by the transducer were translated into the computer via Data Translation Scope, DT9800. This software recorded the sent information and presented them as graphs of pressure fluctuations versus time. For further analysis of the data, the output file of the software was provided to be used in MS Excel spreadsheet programs.

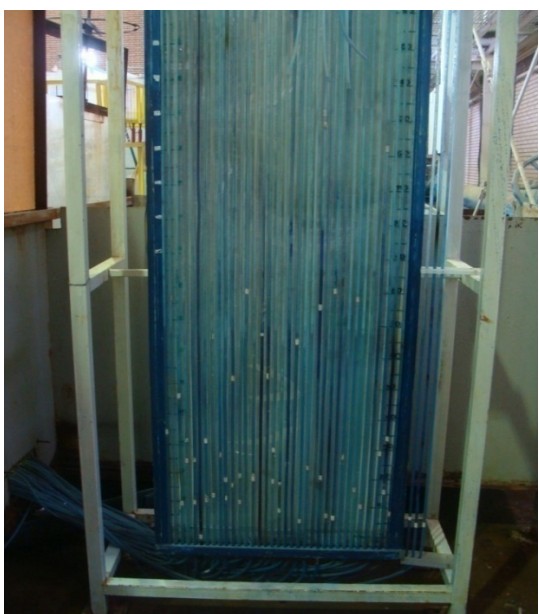

**Figure 3.** The 37 piezometers connected to impact plate for observation of dynamic pressure fluctuations.

## 3. Results and Discussion

### 3.1. Mean Dynamic Pressure Coefficient

The mean dynamic pressure coefficient ($C_P$) was used in the quantitative study of dynamic pressures (Equation (3)). Table 1 contains the mean values of dynamic pressure coefficients at various impact plate angles, plunging depths, and flow discharges.

**Table 1.** Values of mean dynamic pressure coefficients obtained in this research

| α (deg.) | Y (cm) | Q (L/s) 67 | 86 | 161 | 184 |
|---|---|---|---|---|---|
| | | C_P | | | |
| 0 | 0 | 0.360 | 0.386 | 0.812 | 0.858 |
| | 15 | 0.353 | 0.357 | 0.794 | 0.832 |
| | 30 | 0.109 | 0.105 | 0.461 | 0.544 |
| | 45 | 0.057 | 0.054 | 0.318 | 0.438 |
| 30 | 0 | 0.345 | 0.352 | 0.619 | 0.628 |
| | 15 | 0.321 | 0.328 | 0.574 | 0.605 |
| | 30 | 0.078 | 0.135 | 0.459 | 0.429 |
| | 45 | 0.023 | 0.052 | 0.328 | 0.365 |
| 60 | 0 | 0.154 | 0.215 | 0.553 | 0.455 |
| | 15 | 0.167 | 0.205 | 0.514 | 0.432 |
| | 30 | 0.082 | 0.015 | 0.503 | 0.345 |
| | 45 | 0.008 | 0.011 | 0.365 | 0.125 |
| 90 | 0 | 0.114 | 0.191 | 0.379 | 0.168 |
| | 15 | 0.099 | 0.171 | 0.356 | 0.168 |
| | 30 | 0.013 | 0.118 | 0.343 | 0.147 |
| | 45 | 0.012 | 0.077 | 0.136 | 0.044 |

### 3.1.1. Effect of Plunging Depth

Figure 4 shows the mean values of dynamic pressure coefficients measured at the place of jet impingement on the impact plate for different discharges and angles versus the ratio of plunging depth to the jet width ($Y/B_j$).

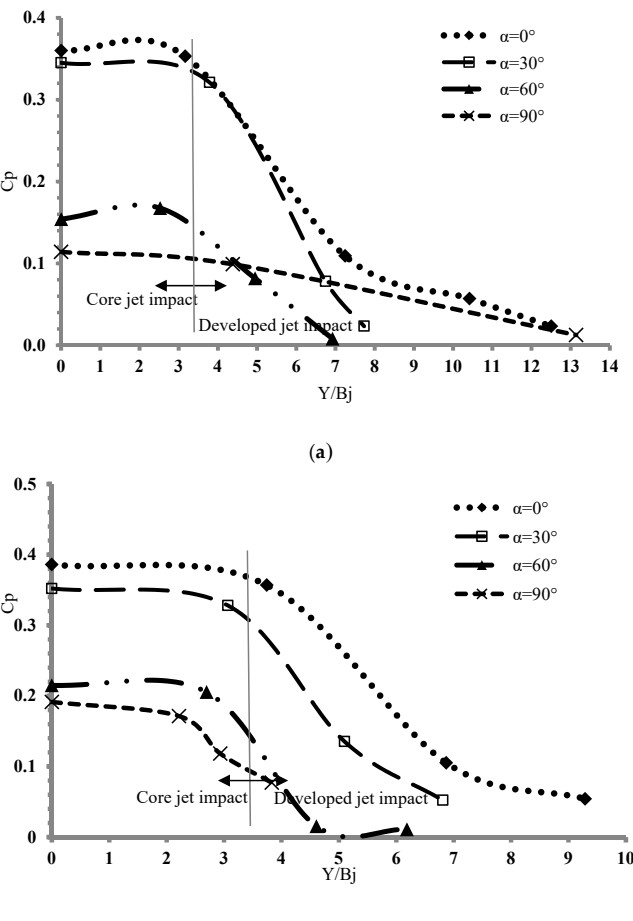

(a)

(b)

**Figure 4.** *Cont.*

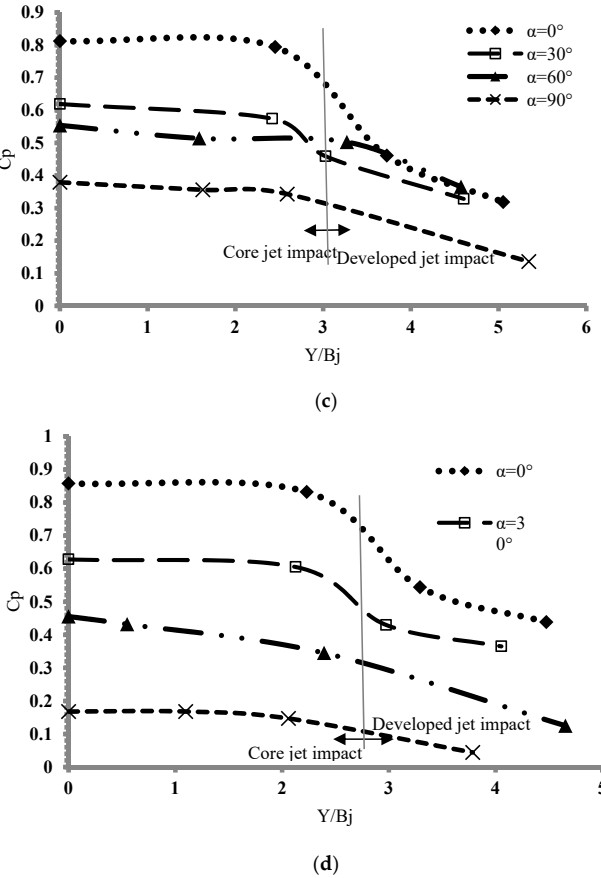

**Figure 4.** Variations of mean dynamic pressure coefficient versus the plunging depth ratio in discharges of (**a**) 67; (**b**) 86; (**c**) 161; and (**d**) 184 L/s.

Figure 4 reveals that for all discharges, when the plunging depth of the pool is zero ($Y/B_j = 0$), the mean dynamic pressure coefficient attains its maximum value at 0°, and its minimum value corresponds to 90°. The reason is that in the case of $\alpha = 0$, the jet impingement is centralized at just one point. By increasing the jet angle, part of the jet is tangent with the impact plate, so the coefficient reduces. Figure 4 also shows that once the plunging depth increases (increment of $Y/B_j$), the value of $C_p$ is at first constant or has no significant reduction, whereas from a certain depth, this coefficient decreases. This occurrence may be argued on the basis of the jet features so that with an increase in the plunging depth, the jet would be of developed type. As regards to the developed jets, they generate more spectral energy at moderate frequencies (100–200 Hz) and low frequencies (less than 20 Hz) compared with the core jets. This is owing to the formation of greater vortices with fewer frequencies in the situation of developed jets [17,18]. The percentage reduction of the mean dynamic pressure coefficient due to the increase in plunging depth is variable from 34% for the 60° and discharge of 161 L/s to 95% for the same angle and with a discharge of 67 L/s.

The impingement of the core jet on the bottom of the downstream pool is a result of the plunge pool being shallow or ineffectiveness of its depth. When the plunge depth is sufficient, the jet impacts the bottom in the non-core or developed nature. In the case of non-core, the dynamic pressure on the bottom decreases, and consequently so does $C_p$. As seen in Figure 4, the value of $C_p$ falls in the decline mode for the range of $2 < Y/B_j < 4$. Therefore, the boundary between the two jet areas, with the core and developed, is within this range. From the region of $Y/B_j > 3$, the plunge value of the downstream pool would be in effect as a result of the jet not impacting in the core state on the impact plate. In accordance with Figure 4a,c, increasing the water depth on the impact plate along with formation of the developed jet brings the diagrams close to each other, which is an indication of the reduction in the effect of impact plate angle. Furthermore, with the increases in discharge, the values of mean dynamic pressure

coefficients increase. Figure 5 illustrates the relation between mean dynamic pressure coefficients and submerge ratio in previous studies compared with the data of this research.

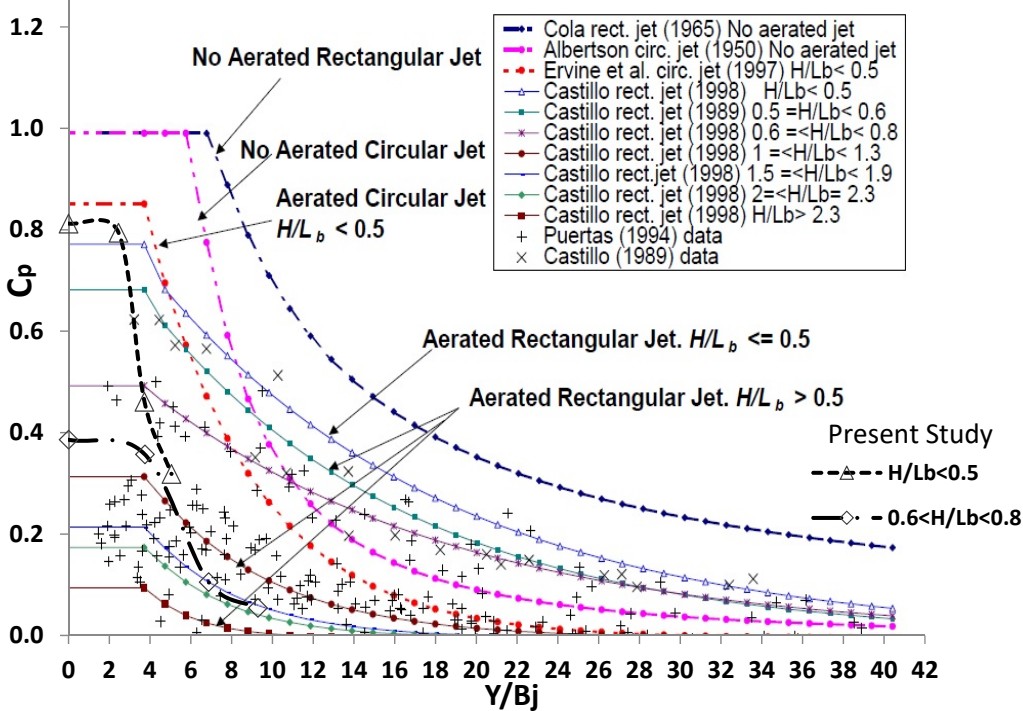

**Figure 5.** Comparison between correlation of $Y/B_j$ and $C_p$ in present research and previous works (reported by Castillo [19]).

Figure 5 shows that in previous research, in agreement with current research, the $C_p$ value decreases from a certain plunge depth onwards. Converting core jet to developed jet, in most investigations, occurs when the ratio of $Y/B_j$ is close to 4. This bound is commonly obtained in this research and some previous works such as Ervine et al. [14] and Castillo [20], and differs, to a degree, from the results of Cola [21] and Albertson et al. [22]. This dissimilarity arises from differences in drop height, jet thickness, and jet shape (aerated and non-aerated, rectangular and circular).

### 3.1.2. Effect of Jet Impact Angle

Figure 6 shows the diagram of $C_p$ versus the impact plate angles in non-plunging depth.

Figure 6 confirms that as the impact plate angle increases, the $C_p$ coefficient decreases. This decrement for the range between 0° and 60° occurs sharper relative to the range between 60° and 90°. An increase in the impact plate angle contributes to a decrease in discharges of 67, 86, 161, and 184 L/s, as almost 74%, 60%, 53%, and 51%, respectively. So, the effect of impact plate angle on the decrease of $C_p$ is more in the lower discharges, and this coefficient increases along with increases in discharge. Hence, it is suggested that in executive works, the maximum discharges of the project design probability maximum flood (PMF) be used for the most critical situations being considered in applying dynamic pressures.

Figure 6 shows the diagram of $C_p$ versus the impact plate angles in non-plunging depth.

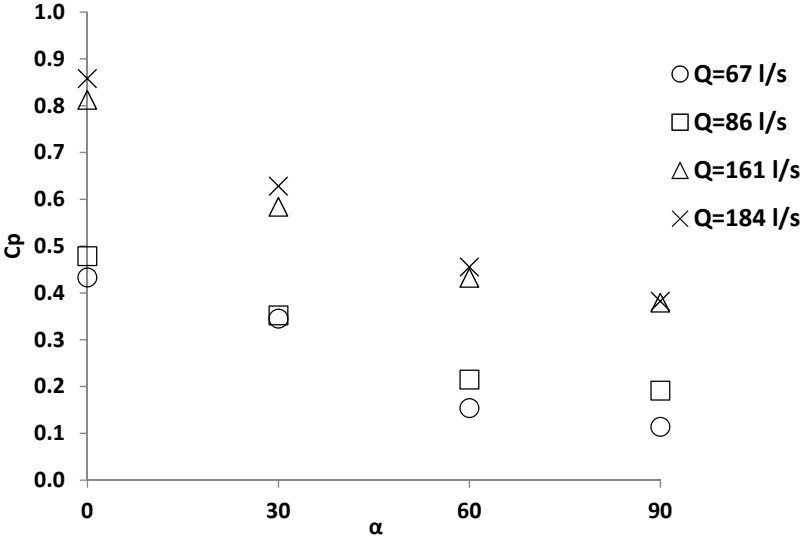

**Figure 6.** Mean dynamic pressure coefficient versus angle of impact plate.

## 3.2. Distribution of Dynamic Pressure on Impact Plate

To study dynamic pressure distribution on $0.5 \times 0.5$ m impact plate in different angles, the changes in $C_p$ values at different radial distances from the impact axis are shown in Figure 7. The figure concerns the discharge of 184 L/s and 0° (with the greatest $C_p$ value) for plunging depths of 0, 15, 30, and 45 cm.

Figure 7a shows the mean dynamic pressure coefficient in the non-plunging depths state. The coefficient value at the center of impact is 0.85, and it decreases away from the center. As seen in Figure 7b, compared with the non-depth case, an increase in the water depth of 15 cm does not have a significant effect on the reduction of $C_p$ values. In fact, the plunging is still of low influence in this stage. At 30 cm depth (Figure 7c), the mean dynamic pressure coefficient is affected by a perceptible decrease. This result indicated that in this situation, the plunging depth was effective in decreasing dynamic pressure. Also, in Figure 7d, in which the plunging depth attained 45 cm, the decline in $C_p$ values was negligible. Thus, the favorable thickness of the plunging depth for the decrease of $C_p$ value happened at the 30 cm depth. An overall consideration of Figure 7 indicated that most dynamic pressures were made in longitudinal distances between 20 and 25 cm, and in transversal distances between 20 and 30 cm. In the cases in which the plunge depth was not so effective, the highest pressure occurred at the center of the impact plate. Any increase in plunging depth causes the center of the effective pressure on the bottom is inclined toward the sides. This may be related to the vortices and turbulent flows created at a slight distance from the point of impact in the direction of the jet central axis [19].

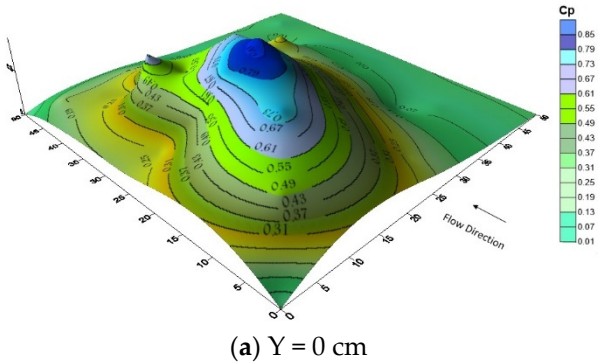

(**a**) Y = 0 cm

**Figure 7.** *Cont.*

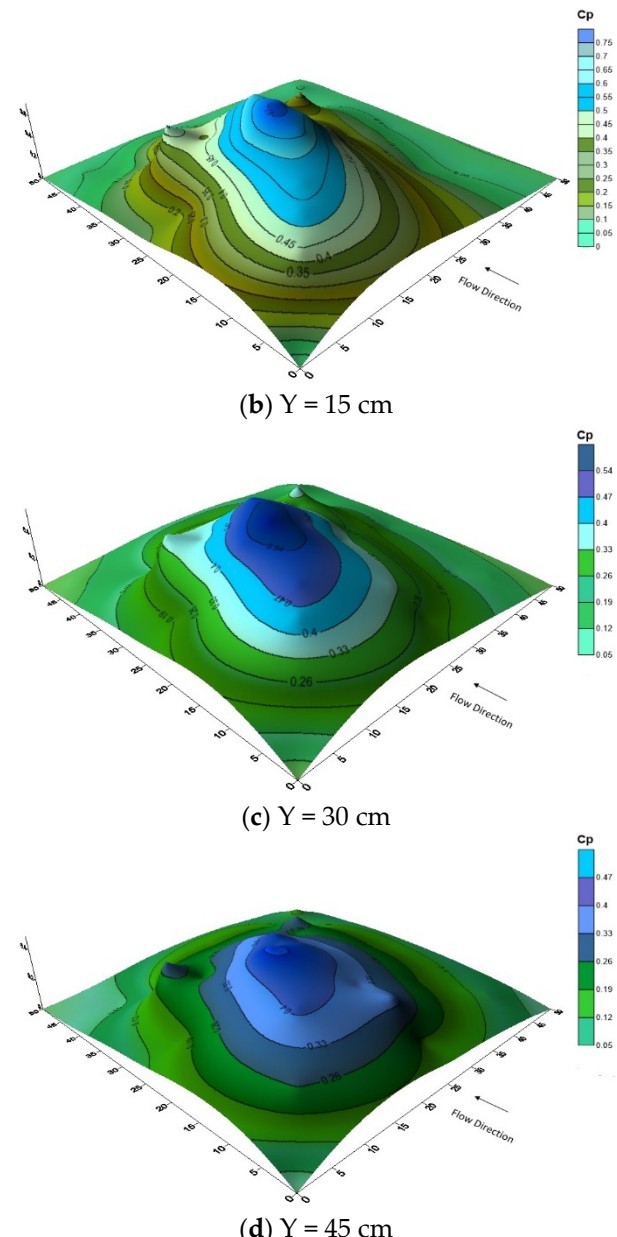

(**b**) Y = 15 cm

(**c**) Y = 30 cm

(**d**) Y = 45 cm

**Figure 7.** Variations of mean dynamic pressure coefficient at radial distances from falling jet axis for discharge 184 L/s at 0° in plunging depths of (**a**) 0; (**b**) 15 cm; (**c**) 30; and (**d**) 45 cm.

### 3.3. Mean Dynamic Pressure Coefficient and Break-Up Length

Experiments were carried out at various depths in the plunge pool to study alterations in the hydrodynamic pressure attributable to variations of the ratio of jet drop length to its break-up length ($H/L_b$). The results are given in Figure 8.

Figure 8 shows that as the impact plate angle relative to the horizontal increases, the value of $C_p$ decreases in all depth situations in the plunge pool. The $C_p$ values at different angles would come close to each other along with an increment in plunging depth. The reason behind this is that when the plunging depth increases, the energy dissipation caused by vortices grows too. Therefore, the effect of the jet impact angle upon dynamic pressure diminishes. Figure 8 confirms that as the $H/L_b$ increases, the coefficient $C_p$ decreases. If the plunging depth is zero or has a low value, the graphs are mostly linear, and the increase in plunging depths causes the graphs to be drawn exponentially. With decreasing plunging depth, the jet drop length increases and motivates the air penetration into the

jet, which in turn brings about energy dissipation. As the plunging depth increases, the jet length, H, would shrink and vortices are produced. Consequently, the energy dissipation of the jet would be affected mainly by the vortices, as a result of which the decreasing trend of the coefficient $C_p$ lessens. Enlargement of H means the extension of the jet drop length and the increase of air entry into the jet subsequently. Thus, it causes more dissipation of energy and decrease in $C_p$ values. Moreover, with a decrease in the break-up length, $L_b$, the jet would be converted from the core to non-core occasion faster, whence a lower dynamic pressure would be exerted [19].

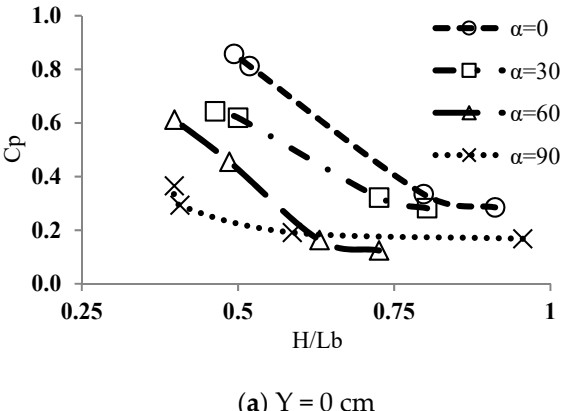

(**a**) Y = 0 cm

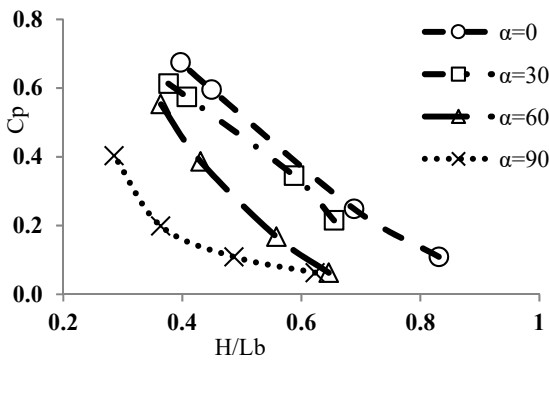

(**b**) Y = 15 cm

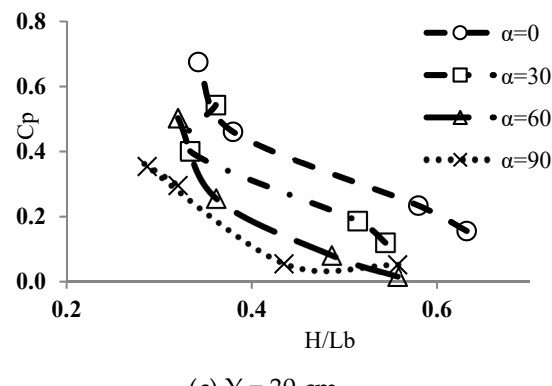

(**c**) Y = 30 cm

**Figure 8.** *Cont.*

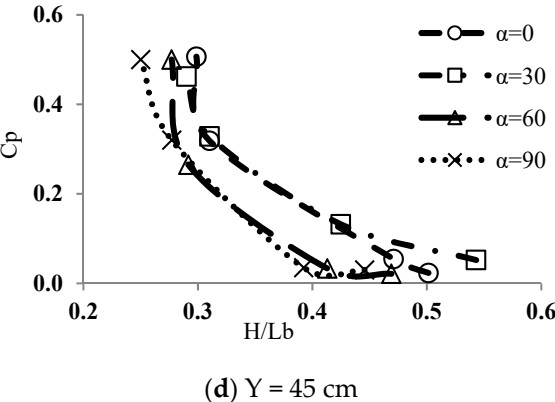

(**d**) Y = 45 cm

**Figure 8.** Mean dynamic pressure coefficient versus break-up length at different impact plate angles and plunging depths: (**a**) Y = 0 cm; (**b**) Y = 15 cm; (**c**) Y = 30 cm; and (**d**) Y = 45 cm.

## 4. Conclusions

In this research, a laboratory model was used to investigate the effect of impingement of a jet after take-off from the fillip buckets. The variables of take-off discharge, jet impact angle, and the downstream plunging depth were simulated. The experiments were carried out both without and with plunging depths of 15, 30, and 45 cm. The most important results obtained from this research are as follows:

- At different discharges, with an increase in plunging depth, the mean dynamic coefficient of the pressure on the impact plate was primarily constant and then decreased.
- As the impact plate angle increased the mean dynamic pressure reduced, this reduction occurred more often from the angles of 0 to 60°.
- The greater the plunging depth, the less the effect of impact plate angle on the mean dynamic pressure was observed.
- A low plunging depth did not give a reduction of dynamic pressure on the walls. This depth was in effect at the time when it reached a certain limit. The bound was obtained as $Y/B_j > 3$ in this research.

**Author Contributions:** M.K.M. and H.H. collected the data and revised original draft preparation, A.A.; provided critical comments in planning this paper, writing and editing the paper, T.M.; M.A.M. read and approved the final manuscript and M.A.M. funded the APC.

**Funding:** This research received no external funding.

**Acknowledgments:** The authors acknowledge Shahid Chamran University of Ahvaz, Iran, for their technical supports.

**Conflicts of Interest:** The authors declare no conflict of interest.

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
