# Peer review of "Physical Modeling of Ski-Jump Spillway to Evaluate Dynamic Pressure"

_water, doi:10.3390/w11081687_

Round 1
Reviewer 1 Report
(a) The authors should explicitly state in the introduction what is novel in this article.
(b) The authors should be aware of the fact that the font size (of texts and equations) should be adjusted.
(c) Please improve Figures-4, 5, and 6.
Author Response
Response to Reviewer 1 Comments
We thank you for providing us the useful and valued comments. In the revised manuscript, we have incorporated all your suggestions and other Reviewers’ comments to improve the quality of the work. Our response to your comments is given below.
(a) The authors should explicitly state in the introduction what is novel in this article.
We thank you the referee for concerning the novelty of the paper. In this research, the entire dominant parameters affecting the pressure at a ski jet were investigated. In addition, apart from most literary works, we used various angles of impact plate. The objectives of the works have been modified as requested by the referee. Kindly see lines 71-73.
(b) The authors should be aware of the fact that the font size (of texts and equations) should be adjusted.
This has been modified in the entire text. Thanks.
(c) Please improve Figures-4, 5, and 6.
Thanks for your useful comments. This has been modified and the Figures (4, 5, and 6) were improved.
Reviewer 2 Report
Measurements of pressures caused by a ski-jump spillway are measured in laboratory experiments. The experiments are carefully conducted, the measurements are detailed and clearly presented, and there is some useful discussion of the results. The paper is well-structured, and the figures are clear and appropriate. Overall this paper would be suitable for publication, but there are a number of minor matters to address first, detailed by line number below.
18 First, a general point: I don’t think the term “sidewall” is appropriate for a surface that is often close to horizontal and I found the term confusing. I suggest it would be better to use something like “impact plate” or “impingement plate” throughout.
19 by ski-jump buckets (delete “the”)
20 depths were used. Discharges of… were chosen.
22 by a transducer.
27 depth varied from
32 pressures resulting
33 scouring [1,2].
35 by a ski-jump
39 on the deflection
44 results show that
45 bubbles disappeared as
46 was reduced… caused by submerged..
49 based on the major effects of
54/5 pressure into dynamic and hydrostatic components
56 time step procedure
58 in a physical
61 results show that
62 fluctuations grow.
79 Adapted from
81 Region A is composed of three sub-regions: A1, A2..
90 turbulence
91 with less energy (delete on downstream)
99 is the acceleration due to gravity (or the gravitational acceleration)
119 It is 75.5 m in height
123 Four discharges of 67,..
124 real discharges of
126 pool were 0, 15,..
151 Piezometers were used to observe the fluctuations of the dynamic pressures on the..
189 part of the jet is
191 or has no significant reduction, while, from a certain depth, this..
199 impingement of the core.. is a result of the plunge pool
202 consequently so does.. As seen in Figure 4,..
208 reduction in the
228 along with increases in discharge.
248 it decreases away from the
251 At 30 cm depth (delete Reaching)
254 decline in Cp
258 the highest pressure
260 This may be related to the
280 relative to the horizontal increases
281 decreases in all depth situations
283 behind this is that
289 and vortices are
294 a lower dynamic
306 The greater the plunging depth
308 A low plunging depth did not give a reduction of
Author Response
Response to Reviewer 2 Comments
Measurements of pressures caused by a ski-jump spillway are measured in laboratory experiments. The experiments are carefully conducted, the measurements are detailed and clearly presented, and there is some useful discussion of the results. The paper is well-structured, and the figures are clear and appropriate. Overall this paper would be suitable for publication, but there are a number of minor matters to address first, detailed by line number below.
We thank you for the encouraging comments. In the revised manuscript, we have attended and incorporated all your suggestions to improve the quality of our work. Following is our response to your comments:
18 First, a general point: I don’t think the term “sidewall” is appropriate for a surface that is often close to horizontal and I found the term confusing. I suggest it would be better to use something like “impact plate” or “impingement plate” throughout.
We change the “sidewall” to “impact plate” entire the manuscript. Thanks for your careful comment.
19 by ski-jump buckets (delete “the”)
Done. Thanks for your accurate comments on the paper presentation. We have modified the sentences as requested.
20 depths were used. Discharges of… were chosen.
Done
22 by a transducer.
Done
27 depth varied from
Done
32 pressures resulting
Done
33 scouring [1,2].
Done
35 by a ski-jump
Done
39 on the deflection
Done
44 results show that
Done
45 bubbles disappeared as
Done
46 was reduced… caused by submerged..
Done
49 based on the major effects of
Done
54/5 pressure into dynamic and hydrostatic components
Done
56 time step procedure
Done
58 in a physical
Done
61 results show that
Done
62 fluctuations grow.
Done
79 Adapted from
Done
81 Region A is composed of three sub-regions: A1, A2..
Done
90 turbulence
Done
91 with less energy (delete on downstream)
Done
99 is the acceleration due to gravity (or the gravitational acceleration)
Done
119 It is 75.5 m in height
Done
123 Four discharges of 67,..
Done
124 real discharges of
Done
126 pool were 0, 15,..
Done
151 Piezometers were used to observe the fluctuations of the dynamic pressures on the..
Done
189 part of the jet is
Done
191 or has no significant reduction, while, from a certain depth, this..
Done
199 impingement of the core.. is a result of the plunge pool
Done
202 consequently so does.. As seen in Figure 4,..
Done
208 reduction in the
Done
228 along with increases in discharge.
Done
248 it decreases away from the
Done
251 At 30 cm depth (delete Reaching)
Done
254 decline in Cp
Done
258 the highest pressure
Done
260 This may be related to the
Done
280 relative to the horizontal increases
Done
281 decreases in all depth situations
Done
283 behind this is that
Done
289 and vortices are
Done
294 a lower dynamic
Done
306 The greater the plunging depth
Done
308 A low plunging depth did not give a reduction of
Done
Reviewer 3 Report
English language should be improved, especially the technical terms should be used in usual sense.
Dynamic pressure is normally given in Pascals, here is in meters. Obviously this is in the „head“ form, then the corresponding physical quantity is the “velocity head” and not “dynamic pressure”.
The physical dimensions are in form “184 ls-1” (line 21), should be “184 l/s” or “184 l.s-1”. Similar in lines 123, 124,…
Line 110, eq. (4) is not the “break-up length”, but the pressure coefficient, the same as (3)!
Line 142, in figure 2 the quantities should be shown clearly, e.g. Bj and points of pressure measurements. The experiment is described not enough in my opinion, the volumetric rate is not the right input parameter for the jet, as velocity is the governing parameter for dynamic pressure.
Generally, all used geometrical quantities should be defined and shown in figures. List (index) of quantities could help.
Line 140: the “side” wall is in reality the “bottom” wall?
Line 150-163 “measurement of dynamic pressure”, the fluctuations of dynamic pressure are not measured properly in my opinion, transducers, tubing, frequency response, etc. are not specified.
In Figure 5 the comparison is not clear, as the presented experiments are in very different regimes than those from literature.
In Figure 7, in graphs the axes are not described.
The paper should be reworked completely, better description, more details should be given and clear figures.
Author Response
Response to Reviewer 3 Comments
Thanks for your observation and comments. We did our best to incorporate all your valued comments. In the review process and based on your comment, we believe the paper has been significantly improved.
English language should be improved; especially the technical terms should be used in usual sense.
Thanks for your comment. We agree with you. In the revised manuscript, we have improved the language and paper presentation, as suggested. Hope our attempt satisfy respected referee.
Dynamic pressure is normally given in Pascal, here is in meters. Obviously this is in the „head“ form, then the corresponding physical quantity is the “velocity head” and not “dynamic pressure”.
We agree with the referee which the dynamic pressure (P) is in Pascal (Pa). However, in the Bernoulli equation, the corresponding terms express as P/γ, which is in meters as is for “velocity head” which mentioned by the referee. In this research, the dynamical pressure coefficient (CP) was used to represent the dynamic pressure which, according to equation 3, is a dimensionless parameter. Thanks for your accurate comments.
The physical dimensions are in form “184 ls-1” (line 21), should be “184 l/s” or “184 l.s-1”. Similar in lines 123, 124,…
We are sorry for this typo error. The dimension has been modified as l/s in the entire text.
Line 110, eq. (4) is not the “break-up length”, but the pressure coefficient, the same as (3)!
Thank you. We have modified this point in the revised manuscript as presented in lines 109 and 110.
Line 142, in figure 2 the quantities should be shown clearly, e.g. Bj and points of pressure measurements.
We modified Fig. 2 in response to your valued comment. The parameter Bj and the location of pressure measurements by piezometers were schematically added in Fig. 2. Thanks for your comment.
The experiment is described not enough in my opinion, the volumetric rate is not the right input parameter for the jet, as velocity is the governing parameter for dynamic pressure.
We agree that the velocity of the jet is an effective and dominant parameter in dynamic pressure, as presented in Equation 3. Moreover, it is well-known that the velocity is descriptive of discharge while the area is a constant value as mentioned in the continuity equation as Q = AUj, where A is flow area (Jet area herein). However, discharge is also an important parameter in the amount of dynamic pressure, as reported in the literature. However, as a finding of this research, according to Fig. 4, at 0°, the maximum pressure coefficient for a discharge of 67 l/s was about 0.35, and for a discharge of 186 l/s, it was about 0.85 which are significantly different.
Generally, all used geometrical quantities should be defined and shown in figures. List (index) of quantities could help.
We thank you for this comment. We have included the variables in Fig. 2a. Regarding the list of variables, we agree that the abbreviation list could help the readers. Since it is not recommended by the journal, it is not incorporated in the manuscript.
Line 140: the “side” wall is in reality the “bottom” wall?
Yes, we agree with you. In the revised manuscript, to overcome this and additional issues raised by other referees, the “side wall” was replaced with “impact plate” throughout the text. .
Line 150-163 “measurement of dynamic pressure”, the fluctuations of dynamic pressure are not measured properly in my opinion, transducers, tubing, frequency response, etc. are not specified.
We agree with you. In the revised manuscript, we have improved the related statements. Kindly see lines152, 157-158.
In Figure 5 the comparison is not clear, as the presented experiments are in very different regimes than those from literature.
We have considered and incorporated this point in the revised manuscript. The graphs are presented in case of H/Lb. Kindly see Fig. 5 which is now consistent with those reported from previous works. Thanks for your valued comments.
In Figure 7, in graphs the axes are not described.
We understand your concern. Therefore, we highlighted the plate dimension in the text as 0.5×0.5 m. The original file is with more resolution which provided enclosed this manuscript.
The paper should be reworked completely, better description, more details should be given and clear figures.
We have incorporated all your comments and other reviewers’ to alter the quality of our works. We believe that your comments improved our manuscript and it is now suitable for publication in the journal. Thanks!
Round 2
Reviewer 1 Report
The authors addressed all my questions and concerns. Thanks a lot! With some of them, I am, however, not yet satisfied requiring another review round. Please improve Figures 4-8.
(a) For figure 4, 5, 6 and 8, please remove the greenish background colour.
(b) For figure 5, please use point & line together to show comparison between present research and previous works. And please explicitly state what is novel here.
(c) Please use rainbow colour scale for figure-7.
(d) Please use point & line together for figure-8.
Author Response
Response to Reviewers
The authors addressed all my questions and concerns. Thanks a lot! With some of them, I am, however, not yet satisfied requiring another review round. Please improve Figures 4-8.
Thanks for your valued comments which improved the contents and presentation of the manuscript.
(a) For figure 4, 5, 6 and 8, please remove the greenish background colour.
We checked the original files and removed any fills and colors as a background for the figures. Kindly see figures 4, 5, 6, and 8. Thank you.
(b) For figure 5, please use point & line together to show comparison between present research and previous works. And please explicitly state what is novel here.
Thanks for your accurate comment. As requested, we used lines and markers for graphs in Figure 5 to be consistent with the charts from other researches. Kindly see the revised version of Figure 5.
As stated in lines 205 -206, the diminishing of the pressure coefficient in this study starts from the range of 2 <Y/Bj <4. In lines 219- 224, the results obtained from Fig. 5 are described and justified.
(c) Please use rainbow colour scale for figure-7.
Done. Kindly see figure 7.
(d) Please use point & line together for figure-8.
Done. Kindly see figure 8.
Reviewer 3 Report
The paper is improved, however still important information is missing.
There is no info about precision of measurements (measuring error) – this is an obligation for any experimental study.
The results in Figure 7 are not described properly. Still, the axes are not assigned (which are streamwise and spanwise directions?). The results are strange, as the distribution should be symmetrical with respect to the flow axis and it is obviously not. What are the small peaks in positions 30,10 and 30,40? Please make a comment.
Author Response
Comments and Suggestions for Authors
The paper is improved; however still important information is missing.
The authors thank the referee for his/her valued comments.
There is no info about precision of measurements (measuring error) – this is an obligation for any experimental study.
The accuracy of the measurements is specified in lines 159-160. Thanks for your accurate comment.
The results in Figure 7 are not described properly. Still, the axes are not assigned (which are streamwise and spanwise directions?). The results are strange, as the distribution should be symmetrical with respect to the flow axis and it is obviously not. What are the small peaks in positions 30,10 and 30,40? Please make a comment.
As the referee correctly stated, the direction of flow was missed in the figure 7. We added the flow direction to recognize the streamwise and spanwise directions of the flow. Moreover, in response to reviewer 1, figure 7 is given in color. Kindly see figure 7.
In case of asymmetry in pressure on the impact plate, depending on the amount of flow and flow rate, the jet position on the impact plate is varying, and the jet may not hit the center of the screen. As stated in manuscript (Lines 196- 199; and lines 260-266) developed jets create both high and low frequency vortices. The formation of such turbulent currents may not cause the maximum pressure to occur in the enclosure. Therefore, the distribution of pressure on the impact plate may not be symmetrical. Also, the formation of these rotational currents may result in the formation of small peaks in different positions on the screen. Few references to support such findings have been cited in the manuscript. Thanks a lot for your efforts to clarify the presentation of the paper.